# Factors Associated with the Knowledge, Attitudes, and Practices of Primary Healthcare Workers Regarding Neglected Tropical Diseases with Skin Manifestations in the Dakar Region, Senegal, 2022

**DOI:** 10.3390/tropicalmed9110267

**Published:** 2024-11-06

**Authors:** Lahla Fall, Karifa Kourouma, Assane Diop, Abdoulaye Diouf, Mohamet Lamine Déthié Sarr, Abdoulaye Dethie Sarr, Hawa Manet, Ivlabèhirè Bertrand Meda, Ndeye Mbacké Kane, Alexandre Delamou, Seni Kouanda

**Affiliations:** 1Order of Malta Hospital, Dakar P.O. Box 11023, Senegal; 2Centre National de Formation et de Recherche en Santé Rurale de Maferinyah (CNFRSR), Forecariah P.O. Box 2649, Guinea; kkourouma@maferinyah.org (K.K.); hawa@maferinyah.org (H.M.); adelamou@cea-pcmt.or (A.D.); 3Public Health Department, Gamal Abdel Nasser University of Gamal Abdel Nasser, Conakry P.O. Box 1017, Guinea; 4Africa Center of Excellence for the Prevention and Control of Communicable Diseases (CEA-PCMT), Faculty of Health Sciences, University of Gamal Abdel Nasser, Conakry P.O. Box 1017, Guinea; 5Department of Dermatology and Venereology, University of Cheikh Anta Diop, Dakar P.O. Box 5005, Senegal; assbindiop@yahoo.fr; 6Statistics and Informatics Division, MANSA LLC, Dakar P.O. Box 15208, Senegal; abdoul.diouf@gmail.com; 7Health Data and Information Management Office-WHO Regional Office for Africa (AFRO), Dakar Emergency Hub, Dakar P.O. Box 4039, Senegal; laminesarr9@gmail.com; 8Department of Mathematics, University of Alioune Diop, Diourbel P.O. Box 30, Senegal; abdoulayedethie.sarr@uadb.edu.sn (A.D.S.); senikouanda@gmail.com (S.K.); 9African Institute of Public Health (IASP), Ouagadougou 12 BP 199, Burkina Faso; medabert@yahoo.fr; 10Programme National de Lutte Contre les MTN (PNL MTN), Ministry of Health and Social Action, Dakar P.O. Box 4024, Senegal; mbackekane2007@yahoo.fr

**Keywords:** knowledge, attitudes, practices, primary healthcare workers, NTDs with skin manifestations, Senegal

## Abstract

Neglected tropical diseases (NTDs) with skin manifestations present a significant health and societal problems challenge worldwide. This study aimed to analyzed factors associated with the knowledge, attitudes and practices (KAPs) of primary healthcare workers (HCW) concerning NTDs with skin manifestations in the Dakar region of Senegal. We conducted a cross-sectional study utilizing a semi-structured questionnaire which was administered to eligible HCW (general practitioners, nurses and midwives) working at the 24 health centers located in the Dakar region. Data were collected using the ODK Collect application (version 2022.3.6). KAPs measures were constructed from questionnaire responses, and bivariate analysis was used to identify associated factors. Of the 187 HCW surveyed, 75.9% were female, with a mean age of 35.1 years (SD: 8.56). The majority of HCWs had a bachelor’s degree (40.6%), were nurses (49.2%) and had between 1- and 10-years of professional experience (52.4%). Regarding their knowledge of NTDs with skin manifestations, only 43.3% of HCW had received initial training on NTDs. The most commonly reported NTDs with skin manifestations were leprosy (53.5%), lymphatic filariasis (51.3%), scabies (49.7%), onchocerciasis (45.5%) and schistosomiasis (42.8%). Television was the main source of information about these NTDs (38.5%). In term of practices, only 47.6% of HCW reported being able to diagnose NTDs with skin. Factors associated with good knowledge regarding NTDs with skin manifestations included having received training on NTDs (*p* = 0.0015) and more years of professional experience (*p* = 0.004). In summary, there is a need to strengthen and raise awareness about NTDs with skin manifestations among HCWs and promote quality management of patients in Senegal.

## 1. Introduction

Neglected tropical diseases (NTDs) pose a significant public health concern due to their extensive human, social, and economic impact as well as the resulting disabilities they can induce. These primarily affect tropical and subtropical regions characterized by limited resources, and where patients often face substantial barriers to access healthcare services [1]. According to the World Health Organization (WHO) in 2016, 20 NTDs were identified as affecting over a billion individual globally and causing more than 500,000 annual deaths [1,2,3]. To overcome this high burden of these diseases, Member States, during the 73rd World Health Assembly, reiterated their commitment to combat NTDs in order to attain the sustainable development goals by 2030. Despite significant progress in the past decade, there are still persistent challenges that pose a threat to the goal of eradicating these diseases. These challenges include timely diagnosis of neglected tropical diseases (NTDs), the accessibility of appropriate medical care and treatment, as well as a lack of reliable and comprehensive data, collaboration, and engagement with stakeholders [1]. Though, previous studies have investigated the knowledge, attitudes and practices of health workers regarding NTDs in Africa [4,5,6,7], only few have focused on those with skin manifestations. These hurdles hinder the overall attainment of the elimination goal, underscoring the need to generate pivotal data to inform national interventions targeting NTDs.

Senegal has 14 endemic NTDs, four of which are primarily skin diseases: leprosy, scabies, cutaneous leishmaniasis, and mycetoma [8]. These diseases account for 15 to 20% of infectious dermatoses in the country’s health facilities [8]. Like other NTD-endemic countries, Senegal aims to eliminate and control NTDs by 2030, in line with the new WHO roadmap. The Ministry of Health and Social Action has established the “Programme National de Lutte contre les MTN” (PNL MTN) to coordinate all specific NTD control programs. Two five-year plans have already been implemented over the past decade (2011–2015 and 2016–2020). By 2020, the coverage targets for various preventive chemotherapies had been achieved in all districts, with average rates of 70%, but the country continues to report NTD cases annually [5].

In 2020, all health districts met the targeted coverage objectives for diseases requiring preventive chemotherapy, with average rates of 70%. That year, 191 new cases of leprosy, 9829 scabies in Koranic schools and prisons, 1424 trichiasis, 608 snakebite envenomation, 104 hydroceles, and 84 lymphedemas of lymphatic filariasis were recorded [8]. Hospital prevalence data, reported by Diop A and colleagues, showed 114 cases of cutaneous leishmaniasis between 2003 and 2020, with a hospital incidence rate of 0.7% [9]. A study of West Africa from 1929 to 2020 documented approximately 2685 cases of mycetoma in 15 countries, with Senegal accounting for 74.1% of cases in the sub-region [5].

These data are underestimated and incomplete due to factors such as the lack of harmonized reporting models in health facilities and late diagnosis owing to low levels of knowledge among healthcare workers. Diagnosing NTDs with cutaneous manifestations is relatively easy, based on observing dermatological lesions, with no mandatory paraclinical investigations required. Senegal, like other endemic countries, prioritizes the fight against NTDs. Consequently, the national program focuses on building healthcare staff capacity. To obtain more reliable data, cases are now reported via the DHIS2 (District Health Information Software 2) platform, a health information management tool used in over 100 countries [10].

Analysing the knowledge, attitudes, and practices of healthcare workers on health issues is crucial to better guide the strategies of national disease control programmes. Several previous studies have investigated the knowledge, attitudes and practices of health workers regarding NTDs in Africa [4,5,6,7], but few have focused on skin-associated NTDs. In Senegal, we could find no research work on the subject. This prompted this study, the aim of which was first to assess the level of knowledge, attitudes, and practices of healthcare workers, and then to analyse the factors associated with knowledge, attitudes and practices concerning skin-associated NTDs in the Dakar region of Senegal in 2022.

## 2. Materials and Methods

### 2.1. Study Settings

#### 2.1.1. General Setting

Senegal, a West African country, had an estimated population of over 17 million in 2021 [11]. The country is divided into 14 administrative regions, including Dakar, the administrative and economic capital. The national health system is structured in a pyramid with three levels: central (Ministry of Health and Social Action), intermediate (14 health or medical regions), and peripheral (79 health districts, 107 health centres, 1499 health posts, and 2268 health huts). In 2022, Senegal had 9744 qualified healthcare personnel, including doctors, nurses, auxiliary nurses, and midwives. Health centres provide medical–surgical, paramedical, outpatient, hospital care, and preventive services.

#### 2.1.2. Specific Setting

Despite covering only 0.28% of the national territory, the Dakar region houses over 25% of Senegal’s population, amounting to more than 4 million people [11]. In 2022, Dakar’s public healthcare system included 14 public hospitals (reference facilities), 4 non-hospital establishments (offering outpatient or day hospitalization services), and 24 health centres. These facilities were staffed by 1412 qualified healthcare professionals: 11% general practitioners, 20% nurses, 23% nursing assistants, and 46% midwives [12,13].

### 2.2. Type and Period of Study

This cross-sectional study was conducted among primary healthcare workers from 31 January to 14 February 2022.

### 2.3. Study Population and Sampling

The sampling frame included all qualified healthcare personnel working in the 24 health centres (N = 1412). Qualified personnel are those involved in care and treatment, with state-recognized training and certification. They include doctors (general practitioners or specialists), midwives, and nurses. The sample size, estimated using the Survey Monkey formula [14], was 172. To account for refusals or non-responses, the sample size was increased by 5%, resulting in a final sample of 189.The questionnaire was randomly administered to available, consenting qualified healthcare staff present on the day of the interview.

### 2.4. Study Variables

#### 2.4.1. Dependent Variables

Two dependent variables were considered:

The level of knowledge about skin-borne NTDs, based on answers to questions on clinical manifestations, transmission modes, risk factors, severity degrees, treatments, and prevention means.

Attitudes and practices towards skin-associated NTDs, based on questions about attitudes towards NTDs, interest in NTDs in healthcare practice, and referral of NTD cases for management.

A correct answer to a single-choice question was coded 1; otherwise, it was 0. For questions with multiple correct answers, the correct answers were averaged, and the question was assigned 1 if the average was ≥50%, and 0 otherwise. The level of knowledge was coded 1 (good) if ≥50% of answers were correct, and 0 (low) otherwise. Similarly, attitude and practice were coded 1 (positive) if ≥50% of answers were correct, and 0 (negative) if not.

#### 2.4.2. Independent Variables (Socio-Demographic)

These variables were studied to determine their association with knowledge, attitudes, and practices regarding skin-borne NTDs:

Age: Four groups: [20–30], [30–40], [40–50], and >50 years.

Sex: Male or female, with the sex ratio calculated based on the number of women (F/M).

Professional experience or Seniority: Number of years in practice, categorized as follows: <1 year, [1–10 years], [10–20 years], and [20–30 years].

Level of education: Highest level attained, categorized as follows: school-leaving certificate, bachelor’s degree, license, master’s degree, and doctorate.

Training on NTDs: Training modules received to reinforce NTD knowledge and skills, categorized as yes or no.

### 2.5. Data Collection

Data were collected using a semi-structured questionnaire designed by the investigator (LF), a dermatology specialist, and validated by other colleagues in the same specialty. The questionnaire was integrated into the ODK Collect software (version 2022.3.6) for data collection.

The survey was conducted by a team of two dermatologists (YS, MDK) and two general practitioners (DDB, SILD), previously trained on the protocol and collection tools under the supervision of the principal investigator (LF). A pre-test was conducted at the Order of Malta Hospital in Dakar with a team of healthcare professionals (sample of 10 healthcare workers) to ensure the relevance and comprehension of the questionnaire.

### 2.6. Data Analysis

Data were extracted from the ODK data collection software and imported into Excel 2019 for analysis.

Results for socio-demographic variables were presented as proportions.

To assess the level of knowledge, the weighted average was calculated to obtain a more precise analysis and therefore a more accurate representation of the data collected.

The results concerning attitudes and practices were presented in the form of proportions.

We studied the relationship between the different variables in our database to determine which ones to cross-reference with our variable of interest. The correlation coefficient, which ranges from −1 to 1, measures the strength and direction of the linear relationship between two variables; for example, age and seniority are positively correlated, with a coefficient of 0.84 indicating a strong positive correlation, which means that in general, as a healthcare worker’s age increases, so does their seniority. This information is crucial to our analysis as it helps us to understand how these variables interact and influences our interpretation of healthcare workers’ knowledge, attitudes, and practices regarding neglected tropical skin diseases.

Since the data were qualitative, we conducted a bivariate analysis to determine the relationship between the variables studied. For quantitative data, Chi-squared tests of independence were performed.

### 2.7. Ethical Aspects

This study was approved by the ethics committee of the Ministry of Health (N: 41 MSAS/CNRES/SP). Authorization from the chief medical officer of the Dakar health region (letter of acceptance N: 95/MSAS/RMD) and health facility managers was obtained. Free informed oral consent from eligible healthcare workers was obtained before each interview.

## 3. Results

### 3.1. Socio-Demographic and Professional Characteristics

Of the nursing staff surveyed, 75.9% were women, 43.9% were aged 30–40, with an average age of 35.1 years. The most common level of education was the baccalaureate (40.6%), with nurses and care assistants being the most common professional categories (49.2%). A total of 52.4% had at least 10 years of professional experience (Table 1).

### 3.2. General Knowledge of NTDs

Regarding NTDs, 53.8% of the surveyed health workers knew the acronym, and 50.3% could provide the WHO definition. When asked to list the NTDs they were familiar with, the most frequently mentioned were leprosy (53.5%), lymphatic filariasis (51.3%), scabies (49.7%), onchocerciasis (45.5%), and schistosomiasis (42.8%). The four NTDs with cutaneous manifestations were also mentioned: leprosy (79.1%), scabies (73.8%), mycetoma (32.1%), and leishmaniasis (19.2%). The results indicated that 43.3% of the health workers had received training on NTDs, with 44.4% having been trained more than 5 years ago. Television was the most frequently cited source of information about NTDs (38.5%) (Table 2).

### 3.3. Specific Knowledge on the Four NTDs with Skin Manifestations

The study revealed that 57.2% of healthcare workers had a good level of knowledge about scabies, particularly regarding its transmission (80.2%) and prevention (64.2%). For leprosy, 36.2% of the staff had a good level of knowledge, especially about its curative treatment (49.7%). In the case of mycetoma, only 14.4% had a good understanding of the disease, with 25% knowing its clinical signs. Lastly, for leishmaniasis, only 5.3% of the nursing staff had a good level of knowledge (Table 3).

### 3.4. Attitudes and Practices of Healthcare Workers towards NTDs with Skin Manifestations

The attitudes and practices of healthcare workers (HCWs) towards NTDs with skin manifestations are shown in Table 4. Regarding attitudes, 29.4% of the healthcare staff surveyed expressed fear of contamination during care, with midwives being the most concerned (38.2%). The most frequently recognized and encountered NTDs with skin manifestations in healthcare practice were, in descending order, scabies (41.2%), leprosy (8%), mycetoma (6.4%), and leishmaniasis (1.1%). In terms of practices, nurses (50.6%), general practitioners (34.8%), and midwives (12.3%) were able to recognize NTDs with skin manifestations. Of these, 62.9% could establish the appropriate treatment and 27.3% referred patients, mainly to dermatologists (66.7%).

### 3.5. Factors Associated with the Knowledge, Attitudes, and Practices Regarding NTDs with Skin Manifestations

In these tables, we cross-tabulated the level of knowledge (Table 5A) and attitudes and practices (Table 5B) of staff with the variables “seniority”, “level of education”, “socio-professional category”, and “initial training received on NTDs”.

## 4. Discussion

Our study population constituted 13.2% of all nursing staff working in the 24 health centres of the Dakar health region. There was a clear predominance of women (sex ratio 3F:1M) and a relatively young population with a majority in the 20–40 age bracket (78.7%). The most representative socio-professional category was nurses (49.2%). The country’s health map for 2022 showed a high proportion of paramedical staff in the Dakar region, with 1524 midwives, 1865 nurses, 2262 nursing assistants (and 267 general practitioners for medical staff) [15] In fact, paramedical staff account for 95% of nursing staff, which could easily explain the predominance of nurses in our study. In Ethiopia, a study of three NTDs also showed that nurses made up the majority of participants (49%). Similar results were obtained by Emeto in Nigeria, with a majority of nursing staff aged between 30 and 39 (37.6%) and a predominance of women (84.1%) [16]. A study carried out in Bangladesh by Kabir H showed a higher frequency in the 26–35 age bracket, with a predominance of women (a third of all professional categories) [17]. In another study carried out in Ethiopia, the average age of participants was 26.4 years, but with a predominance of men (66%) [18].

The most common level of study among healthcare workers was a bachelor’s degree. This result can be explained by the high proportion of nurses and midwives in our study population, and by the fact that diploma training for this professional category lasts three to five years after the baccalaureate. The same observation was made in Bangladesh and Ethiopia, where the “baccalaureate and above” level of education was the most common [17].

The majority of healthcare workers (67.9%) had at most 10 years’ professional experience. This result can be explained by the strong correlation between the young age of the healthcare workers surveyed and seniority. In Senegal, the 15–44 age group represents 74.9% of the working population in urban areas [11]. Emeto in Nigeria found similar results, with 41.3% of healthcare workers having less than five years’ professional experience. In Ethiopia, a study found an average of 7.2 years’ work experience among health workers [18].

The NTDs most known by the staff interviewed were leprosy (53.5%), lymphatic filariasis (51.3%), scabies (49.7%), onchocerciasis (45.5%), and schistosomiasis (42.8%). Leprosy, a “disease of antiquity”, remains a disease known by staff and not necessarily recognized by its clinical signs and treatments. Scabies is raging in epidemic mode in certain centres such as prisons and modern or religious schools due to poor hygiene conditions. This is a frequent reason for consultation and would explain why it is well known by health workers. Schistosomiasis and onchocerciasis are the subject of awareness campaigns and drug distribution in endemic areas. Emeto, in Nigeria, had as a result a level of knowledge mainly on schistosomiasis (78%), rabies (64.5%), onchocerciasis (57.2%), and trachoma (40.7%) [16]. The epidemiological profile of these NTDs varies from one country to another.

The results of the study showed that 43.3% of the health workers surveyed had received training on NTDs. This is most often initial training received during training, or sometimes continuing training. The proportion of training being low should be improved with continuing training sessions, especially since health workers very often change positions after a few years of practice at the level of health structures. Chandler and his colleagues had shown that the diagnostic skills of community health workers improve after training on cutaneous NTDs (15). In Mali, after 12 to 18 months of training on leprosy and common skin diseases for primary health workers, significant improvements were noted in the accuracy of diagnosis of these diseases [19].

Leprosy (79.1%) and scabies (73.8%) are the NTDs best known for their cutaneous manifestations. Indeed, scabies is highly endemic and was recently included in the World Health Organization’s list of NTDs. It is estimated that this parasitic disease affects up to 200 million people worldwide, particularly in low-income populations [20]. Scabies therefore appears to be a more widespread disease from an epidemiological point of view, which may explain this high response rate among health workers. As for leprosy, although it had been the subject of several control strategies with a view to its eradication, with the decline of the disease, it is less and less taught in practice. Health workers very often cite leprosy as an NTD without having a good level of knowledge about the disease. Kabir’s study in Bangladesh showed that the proportion of knowledge of healthcare providers who had received training on leprosy was significantly higher than that of providers who had not.

Mycetoma cases usually present late, at an advanced stage of the disease [1]. In our study, mycetoma was also poorly known among health professionals. This low level of knowledge could be explained by the lack of continuous training of health workers and therefore lead to a delay in the diagnosis of these diseases. Leishmaniasis was the least known NTD in this study. This contrasts with a study reported by Awosan and colleagues in Nigeria which showed a high level of knowledge of this disease by doctors in health facilities [4]. In another study in Sri Lanka, 95.7% of the doctors surveyed had a good level of knowledge about leishmaniasis [21]. These countries are probably highly endemic for this disease and would explain the good knowledge of this parasitic disease. Our results may indicate either a lack of knowledge of leishmaniasis, and therefore underreporting of cases, or a lower prevalence of leishmaniasis in Senegal. The national NTD control programme has only fragmented hospital data. The real epidemiological situation of leishmaniasis is poorly known in Senegal but some geographical areas of endemicity have been identified.

The national NTD control programme has limited hospital data [15,22]. Efforts have been made through the District Health Information Software 2 (DHIS2) platform to harmonise data collection across all health facilities in the country, particularly for skin-transmitted NTDs.

Regarding the attitudes and practices of the health workers surveyed, approximately 30% felt fear of being infected at the time of care, and this was particularly common among paramedical staff (especially midwives). The results showed that apart from scabies, the ability to diagnose other skin-transmitted NTDs was generally limited. The work of Dellar et al. in Ethiopia on three NTDs showed worrying results, with a high level of stigmatisation of health workers, which could be a major barrier to patients’ access to care [18]. In Sudan, 70% of respondents had appropriate attitudes and beliefs about mycetoma, and 49% used good practices to manage the disease [23].

In Senegal, many health programs target nurses to implement their action plans and they are the ones who benefit the most from training and awareness on communicable diseases in general. In the Senegalese health system, they contribute to improving access to primary healthcare [13]. If necessary, patients will be referred to facilities with more specialized medical personnel. In this study, dermatologists were the main referents for the management of NTDs with cutaneous manifestations (66%). Indeed, the Dakar region has nearly 75% of dermatology specialists practicing in public (hospitals and health centres) and private health facilities [24]. Thus, geographical access to dermatologists is relatively easy in the Dakar region.

Regarding factors associated with the knowledge of cutaneous NTDs, occupational category, seniority in the profession, education level of health workers, and initial training received on NTDs were all significant factors, thus positively influencing the level of knowledge. In our study, the overall level of knowledge of these NTDs with manifestation was low. Doctors had better knowledge of the diseases. Given that the survey concerned all health workers, training seems to be the essential element to improve knowledge of NTDs. Emeto and colleagues in Nigeria had noted that prior training on NTDs was a significant predictor of NTD recognition (AOR = 7.09, CI = 3.15–15.93), as was professional experience (AOR = 4.65, CI = 1.20–18.09) [16].

On the other hand, none of the variables studied had a significant effect on the practices and attitudes of health workers. Knowledge generally positively influences attitudes and practices. But the results showed us that the level of knowledge has no influence on attitudes and practices. Since the level of knowledge of health personnel on cutaneous NTDs was low in this study, this result cannot be the subject of a formal conclusion. Thus, this study must be broader and deeper, and other factors must therefore be sought.

All the participants in the study expressed a need for continuing education to update their knowledge of certain rare diseases, such as neglected tropical skin diseases. The COVID-19 pandemic has changed many of our habits, including the development of teleworking and effective, high-quality distance learning. Today, most universities and training institutes have gone back to face-to-face teaching, but some theoretical courses are still taught in the e-learning format (especially in higher education). In practice, therefore, Senegal’s national programme to combat Neglected Tropical Diseases could consider developing distance learning courses, which are less costly, more geographically accessible and can reach a greater number of health professionals.

### 4.1. Limitations of the Study 

There were two main limitations.

First, we conducted a random sample. Care staffs, after identifying their qualifications, were questioned according to their availability at the time of the survey. This could therefore constitute a selection bias.

Secondly, the questionnaire was designed and validated by dermatologists specialized in skin diseases. Thus in the form, questions relating to the knowledge of NTDs were better developed than those concerning attitudes and practices.

### 4.2. Strengths of the Study

This is a pioneering study including all health centres in the Dakar region on the knowledge, attitudes, and practices with respect to the four NTDs with essential skin manifestations.

## 5. Conclusions

NTDs continue to be a real health problem in Senegal, while early diagnosis should be easy. The objective of this study was to assess knowledge about these diseases, as well as the attitudes and practices of qualified health personnel working in health centres in Dakar, and then to identify factors associated with knowledge, attitudes, and practices regarding cutaneous NTDs. The level of knowledge of NTDs with cutaneous manifestations by health workers remains insufficient in general, particularly regarding leishmaniasis and mycetoma. The determining factor associated with knowledge of these diseases seems to be training on these NTDs. However, no factor associated with attitudes and practices was found.

This study allowed us to have an overview of the current situation of knowledge, attitudes and practices of health workers in Dakar, with a view to better guiding strategies to combat these diseases. Senegal has adopted this logic of eliminating NTDs in accordance with the WHO objectives by 2030. More in-depth quantitative, qualitative or mixed studies seem necessary with larger samples in order to achieve the following: (1) have a better overview of the knowledge, attitudes and practices of health workers on these NTDs with cutaneous manifestations (especially in the endemic areas of the country); (2) better understand the interactions and identify associated factors; and (3) act on these factors in order to improve early management, thus reducing the morbidity–mortality rate linked to these NTDs.

## Figures and Tables

**Table 1 tropicalmed-09-00267-t001:** Socio-demographic characteristics of HCWs in the Dakar region, Senegal, 2022 (N = 187).

Variables	Number	%
Age (years)		
20–30	65	34.8
30–40	82	43.9
40–50	30	16.0
>50	10	5.3
Average age = 35.1 (standard deviation = 8.6)		
Gender		
Female	142	75.9
Level of education		
Patent (intermediate school-leaving certificate)	32	17.1
Baccalaureate	22	11.8
License	76	40.6
Master	3	1.6
Doctorate	52	27.8
Not specified	2	1.1
Socio-professional categories		
Nurses/Nursing Assistants	92	49.2
General Practitioners	50	26.7
Midwives	41	21.9
Other * (see below)	4	2.2
Work experience (in years)		
<1 year	29	15.5
1–10 years	98	52.4
11–20 years	44	23.5
>20 years	16	8.6

* Physicians in the process of specialization: infectious disease (2), gynaecology (1), and occupational medicine (1).

**Table 2 tropicalmed-09-00267-t002:** General knowledge about neglected tropical diseases among qualified healthcare workers in Dakar (Senegal) (N = 187).

Variables	Number	%
Knowledge of the NTD acronym (yes)	100	53.8
Correct definition of NTDs according to WHO (Yes)	93	50.3
The NTDs mentioned *		
Leprosy	101	54.6
Lymphatic Filariasis	96	51.3
Scabies	93	49.7
Onchocerciasis	85	45.5
Leishmaniasis	70	37.4
Mycetomas	39	20.8
Schistosomiasis	80	42.8
Rage	40	21.4
Dengue	40	21.4
Snake bite envenomation	8	4.3
Other NTDs **		
Geohelminthiasis	14	7.5
Schistosomiasis	6	3.2
Trachoma	7	3.7
Human African Trypanosomiasis	4	2.1
Previous training on NTDs (yes)	81	43.3
Date of last training on NTDs (n = 81)		
<1 year	11	13.6
1–5 years	19	23.5
≥5 years	36	44.4
NTDs with cutaneous manifestations cited		
Leprosy	148	79.1
Leishmaniasis	36	19.2
Scabies	138	73.8
Mycetomas	60	32.1
Sources of information on NTDs		
Television	72	38.5
Community Outreach Session	49	26.2
Radio	47	25.1
Other ***	17	9.1

* The list of NTDs that was included in the multiple-choice questions. ** Other NTDs” were not included in the questionnaire (series of suggested answers). *** Source of information not specified by the staff surveyed.

**Table 3 tropicalmed-09-00267-t003:** Knowledge of qualified healthcare personnel on the four NTD diseases in Dakar, Senegal 2022 (N = 187).

	Number	%
Knowledge of leprosy		
Clinical signs	11	5.9
Complications of the disease	2	1.1
The mode of transmission	23	12.3
Risk factors for transmission	58	31.0
Management of the disease	93	49.7
Weighted average * = 36.2%		
Knowledge of cutaneous leishmaniasis		
Clinical signs	2	1.1
The mode of transmission	4	2.1
Risk factors for transmission	1	0.5
Management of the disease	0	0
The means of prevention	10	5.3
Weighted average = 5.3%		
Knowledge of scabies		
Clinical signs	42	22.5
The mode of transmission	150	80.2
Risk factors for transmission	33	17.6
Management of the disease	57	30.5
The means of prevention	120	64.2
Weighted average = 57.2%		
Knowledge of mycetoma		
Clinical signs	45	24.0
The mode of transmission	30	16.0
Management of the disease	9	4.8
The means of prevention	31	16.6
Weighted average = 14.4%		

* The different values do not have the same weight overall. Each value has a coefficient noted p (p1 is the weight of the 1st value, p2 is the weight of the 2nd value, etc.). The weighted average is obtained by first multiplying each value by its weighting, and then dividing this result by the sum of the weightings.

**Table 4 tropicalmed-09-00267-t004:** Attitudes and practices of healthcare workers regarding NTDs skin diseases in the Dakar region, Senegal, 2022. (N = 187).

Variables	Number	%
Fear of being contaminated at the time of care (yes)	55	29.4
General practitioners	17	30.9
Nurses	16	29.1
Midwives	21	38.2
Other *	1	1.8
Ability to diagnose NTDs with skin manifestations in healthcare practice (yes) N = 89		
General practitioners	31	34.8
Nurses	45	50.6
Midwives	11	12.3
Other * (see below)	2	2.3
Different NTDs with skin manifestations encountered in healthcare practice N = 187		
Leprosy	15	8.0
Leishmaniasis	2	1.1
Scabies	77	41.2
Mycetomas	12	6.4
Others NTDs **	10	5.3
Cutaneous NTD case management capacity and treatment compliance (yes) N = 89	56	62.9
Referral of NTDs with skin manifestations (yes)	51	27.3
Nurse focal point (district)	4	7.8
Regional focal point	0	0
General practitioners (district or regional)	5	9.8
Dermatologists	34	66.7
Other ***	7	13.7
No answer	1	2

* Physicians in the process of specialization: infectious diseases (2), gynaecology (1), and occupational medicine (1). ** Other NTDs (other than NTDs with cutaneous manifestations) not included in the questionnaire but known to and managed by the healthcare workers questioned. *** Other referents different from the proposed answers.

**Table 5 tropicalmed-09-00267-t005:** (**A**) Factors associated with the knowledge of healthcare workers (HCWs) towards NTD diseases with skin manifestations in the Dakar region, Senegal, 2022. (**B**) Factors associated with attitudes and practices of healthcare workers (HCWs) in health centres towards NTD diseases with skin manifestations in the Dakar region, Senegal, 2022. (**C**) Contingency table between knowledge and practical attitudes of healthcare workers.

(A)				
Variables	Modalities	Knowledge Level		*p*-Value
		Good	Low	
Level studies	Patent	0	33	3.54 × 10^−12^
	Baccalaureate	1	21	
	License	1	75	
	Master	1	2	
	Doctorate	24	29	
Socio-professional categories	Nurses	2	91	2.27 × 10^−11^
	General Practitioners	24	29	
	Midwifes	1	40	
Initial training on NTDs	No	9	97	0.015
	Yes	18	63	
Seniority	Less than 1 year	10	19	0.004
	01–10	15	83	
	11–20	1	43	
	21–30	1	14	
	More than 30 years	0	1	
**(B)**				
**Variables**	**Modalities**	**Attitudes and Practices**		***p*-Value**
		**Positive**	**Negative**	
Level of education	Patent	11	21	0.1926
	Baccalaureate	8	14	
	License	20	56	
	Master	1	2	
	Doctorate	23	29	
Socio-professional categories	Nurses	30	62	0.09219
	General Practitioner	22	2	
	Midwives	10	31	
Seniority	Less than 1 year	0	1	0.1509
	1–10	32	65	
	11–20	11	33	
	21–30	9	6	
	More than 30 years	12	18	
Initial training on NTDs	Yes	23	58	0.1891
	No	41	65	
**(C)**				
**Variables**	**Modalities**	**Attitudes and Practices**		***p*-Value**
		**Positive**	**Negative**	
level of knowledge	Good	13	14	0.1529
	Low	51	109	

(**A**) The results indicate that health workers with a level of education equal to “Doctorate” have a better level of knowledge about NTDs than those with “Brevet”, “Baccalaureate”, “License”, and “Master”. This relationship is statistically significant with a *p*-value of 3.54 × 10^−12^. With regard to the functions of the healthcare workers questioned, the “nurses” and “midwives” categories had little knowledge of NTDs. General practitioners had a higher proportion of good knowledge, with a *p*-value of 2.27 × 10^−11^. Initial training on NTDs also had a statistically significant relationship with the level of knowledge (*p*-value at 0.015). Healthcare workers who did not receive any initial training on NTDs mainly had low levels of knowledge, unlike those who did. Length of service was also an influential factor (*p*-value 0.004). Those with between 1 and 10 years’ professional experience had a higher proportion of good knowledge than the other groups. (**B**) The results show no relationship between attitudes and practices and the independent variables studied. (**C**) There was also no statistically significant relationship between the level of knowledge and practical attitudes in this study (*p*-value of 0.1529).

## Data Availability

The database used in this study is available upon request from the corresponding author.

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
