# Peer review of "Factors Associated with the Knowledge, Attitudes, and Practices of Primary Healthcare Workers Regarding Neglected Tropical Diseases with Skin Manifestations in the Dakar Region, Senegal, 2022"

_tropicalmed, 2024, doi:10.3390/tropicalmed9110267_

Round 1
Reviewer 1 Report
Comments and Suggestions for Authors
The objective of this study was to analyse the factors associated with the knowledge, attitudes, and practices of qualified primary health care workers regarding NTDs with skin manifestations in the Dakar region of Senegal in 2022. 187 nurses underwent a kind of exam about NTDs that manifest on the skin. Unsurprisingly, the more common diseases were recognised and respondents knew more about them. The only factor influencing the results was the extent of previous training. This will be like that everywhere in the world and is simply human.
I would recommend to shorten the paper (we asked/they answered/these are the results) and to focus more. There is a lot of information, albeit interesting, about senegalese health system, NTDs in general an so on, which however distracts from the actual statement. The tables can be moved to an annex.
The basic message for me seems to be, that there is a lack of training, that can be dangerous for the patients. In one sentence it is mentioned, that more training has proven to be helpful (Line 333, 334). Can this be explained further? What is suggested to improve the situation? What could be incentives for training? What type of training would be preferred (in presence, online, compulsory,...) Maybe you could do a follow-up of your questionaire to look at that.
Author Response
Response to Reviewer #1:
The objective of this study was to analyze the factors associated with the knowledge, attitudes and practices of qualified primary health workers regarding NTDs with cutaneous manifestations in the Dakar region of Senegal in 2022. 187 nurses underwent some kind of examination on NTDs with cutaneous manifestations. Not surprisingly, the most common illnesses were recognized and respondents knew more about them. The only factor influencing the results was the extent of previous training. It will be like this everywhere in the world and it is simply human.
The objectives of this study have been explained in more detail in the introduction:
Senegal, like other endemic countries, has set itself the goal of eliminating and controlling NTDs by 2030, in accordance with the WHO road map, and has made progress fight against NTDs a priority area of ​​action. Despite the numerous efforts of the State and its technical and financial partners, difficulties persist and delay the achievement of the targeted objectives.
In this study, we looked at the knowledge, attitudes and practices of qualified healthcare workers concerning NTDs, and in particular cutaneous NTDs. Analysis of the level of knowledge and behaviors of healthcare workers on health issues is necessary for a better orientation of the response. Several studies have examined the knowledge, attitudes and practices of primary health care workers regarding NTDs in Africa [4, 5, 6, 7], but few have focused on NTDs with cutaneous manifestations. The aim of our study was therefore to analyze the factors associated with the knowledge, attitudes and practices of qualified health workers regarding the four skin-associated NTDs in the Dakar region of Senegal in 2022.
I would recommend to shorten the paper (we asked/they answered/these are the results) and to focus more. There is a lot of information, albeit interesting, about senegalese health system, NTDs in general an so on, which however distracts from the actual statement. The tables can be moved to an annex.
In this document, we have chosen to provide as much information as possible about the Senegalese health system, and to assess knowledge about NTDs in general and cutaneous NTDs in particular. The tables have not been appended for a better understanding of the results. The tables have been integrated into the text in accordance with the journal's instructions.
The basic message for me seems to be, that there is a lack of training, that can be dangerous for the patients. In one sentence it is mentioned, that more training has proven to be helpful (Line 333, 334). Can this be explained further? What is suggested to improve the situation? What could be incentives for training? What type of training would be preferred (in presence, online, compulsory,...) Maybe you could do a follow-up of your questionaire to look at that.
We have taken on board the comments and suggestions and reworded this part of the discussion:
In general, infectious diseases, including neglected tropical diseases, are taught in theory during the initial training of health professionals. As they move into professional practice, these workers need to continue their training in order to improve their skills or acquire new knowledge about the diseases. This lack of continuing education could also explain the delay in diagnosing these diseases.
Always in discussion:
All study participants expressed a need for continuing education to update their knowledge on certain rare diseases, such as neglected tropical dermatoses. The COVID 19 pandemic has disrupted many of our habits, notably the development of teleworking and effective, quality distance learning. Today, most universities and training institutes have returned to face-to-face teaching, but certain theoretical courses are still taught in e-learning format (particularly in higher education). In practice therefore, the national program to combat neglected tropical diseases in Senegal could consider developing distance training, which is less expensive, more geographically accessible and which reaches a larger number of health professionals.

Reviewer 2 Report
Comments and Suggestions for Authors
The topic of the study by Lahla Fall and co-workers is interesting and relevant. Overall, the work is nicely presented, but there are a number of flaws that should be given attention before the manuscript is acceptable for publication in Tropical Medicine and Infectious Disease.
Abstract
Line 35: specify ‘healthcare personnel’.
Line 37: nurses = 49.2%. What is the rest? And 52.4 of the total number of participants, or of the share of nurses?
Lines 39-41 should be written more clearly.
Line 41; greater than what?
Line 43: what do you mean with ‘qualified health care personnel’? In other words, what is unqualified in this context? See also section 2.3, line 143.
Line 44: what do you mean with ‘a correct attitude…’? Diagnosis, treatment?
Line 45: ‘no associated factors’: such as?
In general, be more consistent when using terms and avoid vagueness.
Introduction
Line 61: by whom of which body were the goals set?
Last paragraph: the aim of the study should be rewritten. It is too vague now. Also, connect with the last paragraph of the discussion (which can also be clearer). Lines 418-421 in the conclusion is the best of the three. Please align this better. Line 95: name the NTDs here.
Materials and Methods
2.1.: specify ‘primary staff care’ and make this uniform throughout the manuscript (compare for instance the abstract).
2.2.2.: add a date from which the figures presented come.
2.3.: add informed consent.
2.5.: provide initials of the persons mentioned if they are author of the manuscript (investigator, line 205; principal investigator, line 212, etc.)
2.6. this section seems incomplete; see for instance Table 5 for which statistical methods were used that are not mentioned or described in M&M.
Results
Data presented in tables: replace comma for decimals by decimal point.
Describe clearer what the columns of the tables include (Table 1, the headings ‘sample’, ‘improved sample’, for instance and check other tables as well.).
3.2.: means are given, but probably a median value, min and max, interquartile range are relevant as well.
Table 2: ‘(standard deviation) = 8-56’ is unclear. Under level of education: patent? …years old, do you mean years in function, employed?
Table 4: mention de four diseases in the legend. Add N=187.
Table 5: Other* (see below) and Others** (in French) ask for a better clarification.
Lines 281-284: this paragraph is not clear. What, for instance is meant with ‘All age groups had a p-value above 5%’? Also, make sure that M&M contains all information about what was compared to what, what was tested, H0, etc.
Tables 6A/B: b should be clarified, also in M&M section.
Please explain the relevance of the 95% CI. What do they mean? Explain the extremely low and extremely high values that were calculated. Is this reality? Also, explain NA.
Discussion
In the discussion, too many results are repeated (e.g., many figures). Please avoid or at least reduce to only what is really relevant for the discussion. See, for instance, lines 325-329, line 338.
Line 402-403: explain and connect better with the aim of the study.
Limitations of the study
What exactly is paramedical staff? This term was not used so far in the manuscript. Be consistent with terminology.
Second… (lines 409-411) needs to be clarified.
Not only mention limitations, but also strengths of the study.
Conclusions
Also pay attention to possible future action points to improve the situation around knowledge. Any policy that can be followed? Recommendations to the health authorities in Senegal?
References
Quite a number of refences in the list seems incomplete regarding bibliographical data. Also add the date on which websites were accessed.
Author Response
Response to Reviewer #2:
Abstract
Line 35: specify “health personnel”.
Line 37: nurses = 49.2%. What's the rest? And 52.4 of the total number of participants, or from the nurses?
Lines 39-41 should be written more clearly.
Line 41; bigger than what?
Line 43: What do you mean by “qualified health personnel”? In other words, what do we mean by “unqualified” in this context? See also section 2.3, line 143.
Line 44: What do you mean by “a correct attitude…”? Diagnosis, treatment?
Line 45: “no associated factors”: such as ?
In general, be more consistent in your use of terms and avoid inaccuracies.
We have taken into account the comments and suggestions made in the summary. We have reworded the entire summary section to make it easier for readers to understand the study.
Abstract:
Neglected tropical diseases (NTDs) with cutaneous manifestations constitute a heavy health and social burden. The objective of this study was to analyze the factors associated with the knowledge, attitudes and practices of qualified primary health workers regarding these skin diseases in the Dakar region of Senegal in 2022. This was a cross-sectional analytical study in care environment. A random sampling technique was used for all 24 health centers in the Dakar region. A semi-structured questionnaire administered with the ODK Collect application (version 2022.3.6) was used to collect information on sociodemographic characteristics and knowledge, attitudes and practices regarding cutaneous NTDs.
Our study population consisted of 187 qualified health personnel working in health centers in the Dakar medical region. These are competent, trained personnel providing care and regulated in accordance with national and international standards.
French A clear female predominance was found (sex ratio 3F: 1H) with an average age of 35.1 years and a bachelor's degree (40.6%). Nurses represented 49.2% of the health personnel surveyed, the remainder being general practitioners (26.7%) and midwives (21.9%). Of this study population, 52.4% had been practicing for at least 10 years. Previous training on cutaneous NTDs was noted in 43.3%. The NTDs most often cited by qualified health personnel were leprosy (53.5%), lymphatic filariasis (51.3%), scabies (49.7%), onchocerciasis (45.5%). and schistosomiasis (42.8%). Scabies and leprosy are the best-known NTDs with cutaneous manifestations. Mycetoma and leishmaniasis are virtually unknown. Among skilled healthcare workers, an attitude is a tendency to make a judgment or take an action when faced with a patient or illness, while a practice is the action or reaction in response to the given situation. Nurses (50.6%) had a greater capacity to diagnose and manage NTDs with cutaneous manifestations. Referral for the management of these diseases was mainly towards dermatologists (66.7%). Nearly 70% of qualified healthcare personnel had a correct attitude towards these infectious diseases.
After analysis, the only factor associated with knowledge of cutaneous NTDs was the training received. No associated factors regarding attitudes and practices of skilled health workers regarding cutaneous NTDs were found in this study.
Introduction
Line 61: by whom and by what organization were the objectives set?
We have taken into account the comments and suggestions and have reworded this part of the introduction:
However, considerable progress has been made over the past decade. For example, at least one NTD has been eliminated in 42 countries, territories and regions, and more than 600 million people no longer need interventions for multiple NTDs. However, major challenges remain and threaten the achievement of elimination targets. At the 73rd World Health Assembly, Member States renewed their commitment to tackling neglected tropical diseases to achieve the Sustainable Development Goals set out in the new Roadmap 2021-2030. New challenges include early diagnosis of NTDs, monitoring and evaluation of actions in endemic countries, access to medical care and treatment, combating marginalization and advocacy with partners and stakeholders for the next decade [1].
Last paragraph: the objective of the study should be rewritten. It's too vague now. Additionally, link to the last paragraph of the discussion (which may also be clearer). Lines 418-421 of the conclusion are the best of the three. Please
Round 2
Reviewer 1 Report
Comments and Suggestions for Authors
You succeeded in making some things clearer, but I still consider the text too long and not concise enough. A scientific article focuses on the scientific question and is usually not meant to give an overview of all topics related to it; unless you write a review. Please change from a narrative to a scientific article and your message will be much clearer and supported by data.
I´m also wondering about the statistical analysis. NOw that I read it again, I´m in severe doubt about those confidence intervalls in Table 6. They dont make sense. Table 6 (A & B) is not self-explanatory or clear. In fact, I didn´t understand how you generated those numbers, although you mentioned the statistical tests you used. You should check table 6 (A & B) again thoroughly. Make sure you used the appropriate statistical test, rewrite it or leave it out.
Author Response
Response to Reviewer #1:
First of all, on behalf of the co-authors and myself, I would like to thank you warmly for your thorough reading of our manuscript, and for your very pertinent comments and suggestions.
Here are the responses
In the send round of review, the reviewers', comments are as below:
You succeeded in making some things clearer, but I still consider the
text too long and not concise enough. A scientific article focuses on
the scientific question and is usually not meant to give an overview of
all topics related to it; unless you write a review. Please change from
a narrative to a scientific article and your message will be much
clearer and supported by data.
The entire document was revised and rewritten to make it a scientific article, with the objectives of the study set out more precisely. The study framework was simplified. Some tables have been eliminated (distribution of workforce by health district) and the "place of work" variable has also been removed.
I´m also wondering about the statistical analysis. NOw that I read it
again, I´m in severe doubt about those confidence intervalls in Table
6. They dont make sense. Table 6 (A & B) is not self-explanatory or
clear. In fact, I didn´t understand how you generated those numbers,
although you mentioned the statistical tests you used. You should check
table 6 (A & B) again thoroughly. Make sure you used the appropriate
statistical test, rewrite it or leave it out.
In this table, we cross-tabulated the "knowledge" variable with the "age group", "seniority", "educational level" and "socio-professional category" variables.
In order not to rely solely on a point estimate, we considered it necessary to estimate the parameters using a 95% confidence interval.
the lower bound of an CI is obtained by
and the upper bound by
If we find NA = Not Applicable, this means that the parameter is not significant because the is too large when tending towards infinity.
For infinities, this quantity is too close to the parameter p that we are trying to estimate. We estimate that the population follows a normal distribution, so we take which is 1.96.
To measure the significance of our IC95% confidence interval, we calculated the p-value to ensure that the confidence interval is not statistically null.
A p-value greater than 5% means that our IC95 is not significant, in other words that the estimated parameter is not statistically significant. In other words, the estimated parameter is not statistically significant, meaning that it is zero and therefore cannot be used to explain the agents' knowledge.
Beta is the probability of rejecting the null hypothesis.
I did not have enough time for a working meeting with all the statisticians involved in this study. If tables 6A and 6B are still incomprehensible, we will change our statistical methods as soon as possible.
Round 3
Reviewer 1 Report
Comments and Suggestions for Authors
I think with some small corrections and some adjustments to the statistics the paper will be fine for publication

Author Response
First of all, on behalf of the co-authors and myself, I would like to thank you for the thorough reading of our manuscript, as well as for your very relevant comments and suggestions.
The entire document has been revised and rewritten for better understanding. We have also noted in your remarks, an English problem. Here are the point-by-point responses to the comments. Please also find a "track changes" and a clean copy of our manuscript.
Line 30-31 Check Sentence, looks like a word is missing
Line 38-41 Check English
Line 47 Leave SD out, not necessary
Line 60 Fill brackets “the majority (XX)”
Line 66 Number is missing (How many times more likely?)
Line 67 I am still not sure about those confidence intervals. The range is huge.
Line 66 – 67 There is a sudden jump to the conclusion. Rephrase to make the flow of thought more
coherent.
Abstract reformulated
Line 146 – 149 almost the same sentence as in 161- 162 Leave out or rephrase
Line 163 – 166 ‘Don’t cut out “cases of”
reworded paragraph
Line 182 Check the wording, is underestimated the right expression?
Line 185-187 This sentence seems out of context here. Put it in a better place creating a good transition
reworded paragraph
Line 225 Make sure you haven’t lost half of the sentence
reworded paragraph
Line 242 Reference? Or what does XX [KK6] mean?
reworded paragraph
Line 296 – 297 Check wording, sentence structure and Reference
Line 302 Reference? Repetition 2 x obtained.
Line 305 – 319 Content seems to be like Line 318- 328
Line 332 – 345 vs 347-378 Which one? Line 332 – 345 is easier to understand
reworded paragraph
Line 397 Check phrase
Line 410- 425 Which part of the text do you keep?
Line 491 – 505 A confidence interval says that there is a 95 % probability that a result is within the
recorded range. When the values reach in the infinite range, this does not make any sense. Please
take these tables out and find a more appropriate way to show the connection between knowledge
and training
reworded paragraph
Line 507 – 524 The results mentioned can be shifted to Results and in Discussion related to other
countries (Example: “Our study population was comparable to other countries regarding sex, age and education.” Then mention the results of other surveys
Separate results more clearly from discussion. Take the essence of the results and discuss them with other findings.
reworded paragraph
Insert conclusion: Example “Early detection is of great importance for the control and eradication of
neglected tropical diseases. Especially in the case of neglected tropical diseases that manifest
themselves on the skin, the so-called visual diagnosis is an important tool. Healthcare workers must
be regularly trained and educ
conclusion reformulated
